# Automatic hoof-on and -off detection in horses using hoof-mounted inertial measurement unit sensors

M. Tijssen[1]*, E. Hernlund[2], M. Rhodin[2], S. Bosch[3,4], J. P. Voskamp[5,6], M. Nielen[1], F. M. Serra Bragança[6]

1 Department Population Health Sciences, Faculty of Veterinary Medicine, Utrecht University, Utrecht, The Netherlands, 2 Department of Anatomy, Physiology and Biochemistry, Swedish University of Agricultural Sciences, Uppsala, Sweden, 3 Inertia Technology B.V., Enschede, The Netherlands, 4 Department of Computer Science, Pervasive Systems Group, University of Twente, Enschede, The Netherlands, 5 Rosmark Consultancy, Wekerom, The Netherlands, 6 Department Clinical Sciences, Faculty of Veterinary Medicine, Utrecht University, Utrecht, The Netherlands

* m.tijssen@uu.nl

**Data Availability Statement:** All relevant data are within the paper and its Supporting Information files.

## Abstract

For gait classification, hoof-on and hoof-off events are fundamental locomotion characteristics of interest. These events can be measured with inertial measurement units (IMUs) which measure the acceleration and angular velocity in three directions. The aim of this study was to present two algorithms for automatic detection of hoof-events from the acceleration and angular velocity signals measured by hoof-mounted IMUs in walk and trot on a hard surface. Seven Warmblood horses were equipped with two wireless IMUs, which were attached to the lateral wall of the right front (RF) and hind (RH) hooves. Horses were walked and trotted on a lead over a force plate for internal validation. The agreement between the algorithms for the acceleration and angular velocity signals with the force plate was evaluated by Bland Altman analysis and linear mixed model analysis. These analyses were performed for both hoof-on and hoof-off detection and for both algorithms separately. For the hoof-on detection, the angular velocity algorithm was the most accurate with an accuracy between 2.39 and 12.22 ms and a precision of around 13.80 ms, depending on gait and hoof. For hoof-off detection, the acceleration algorithm was the most accurate with an accuracy of 3.20 ms and precision of 6.39 ms, independent of gait and hoof. These algorithms look highly promising for gait classification purposes although the applicability of these algorithms should be investigated under different circumstances, such as different surfaces and different hoof trimming conditions.

## Introduction

Gait analysis is an important element for the understanding of equestrian sport and can be performed by examining gait characteristics and body segment positions of the horse while moving. Gaits can be distinguished by their foot-fall pattern in addition to knowledge about the duration of the support phase compared to the whole stride duration of one leg [1]. For

**Funding:** No external funding was utilized for the current analysis of the existing data. Indirect support was provided through salaries by the home institutions of all co-authors. Inertia-Technology B.V. provided support in the form of salary for author S. Bosch, Rosmark Consultancy provided support in the form of salary for author J. P. Voskamp, the specific roles of these authors are articulated in the 'author contribution' section. The funders did not have any additional role in the study design, data collection and analysis, decision to publish, or preparation of the manuscript.

**Competing interests:** The authors have read the journal's policy and the authors of this manuscript have the following competing interests: S. Bosch is a paid employee of Inertia-Technology B.V., the company that sells the inertial sensor system used for data collection, and has received salary support for his role in this study. J.P. Voskamp is founder of Rosmark Consultancy and has received salary support for his role in this study. Inertia-Technology B.V., Rosmark Consultancy and Utrecht University are partners in the EquiMoves® corporation. This does not alter our adherence to PLOS ONE policies on sharing data and materials.

gait classification, the fundamental locomotion characteristic is the timing of hoof placement, i.e. hoof-on and hoof-off events from all limbs. These events can be examined visually but due to the limitations of the temporal resolution of the human eye [2] there are limitations to how well these events can be distinguished. Instead, objective measurement tools such as force plates, optical motion capture (OMC) systems and inertial measurement units (IMUs) are used [3].

In general, the force plate is considered the gold standard for kinetic gait analysis. With this method, hoof impacts can be registered from the vertical force signal by applying a threshold which is subjective. Furthermore, data collection is time consuming [4] and this method can only be used in a laboratory settings. In addition, multiple consecutive strides can only be measured with a force measuring treadmill and by force measuring shoes [3], which can alter the kinematics [5, 6]. OMC systems and IMUs can also be used to measure consecutive strides and OMC systems are considered the gold standard for kinematic gait analysis. However, these systems are expensive and not easy to relocate due to the significant number of cameras and infrastructure needed. Therefore, OMC systems have limited usefulness in field conditions [7]. IMUs can easily be used in field conditions because they are portable, wireless and are becoming relatively cheap. Consequently, IMUs improve the possibilities for gait analysis in field conditions.

Previous studies investigated the accuracy and precision of IMUs compared with the force plate [8, 9] and OMC systems [7, 10–12] and showed the potential of IMUs for gait analysis and classification. However, analysis of the data and extraction of hoof-events was performed manually or semi-manually which is time consuming and subjective. Time reduction and objectivity can be gained by developing an algorithm for automatic detection of hoof-on and hoof-off events from the output of the IMUs [3]. The output of the IMUs consists of tri-axial acceleration and angular velocity signals. Recently, one study was performed to evaluate multiple algorithms for hoof-event detection and validation against the force plate [13]. In this study, distal limb mounted IMUs were used and the best performing algorithm of this study showed an accuracy between -19.7 and 17.6 ms and a precision between 7.5 and 31.0 ms, depending on gait, limb and hoof-event [13]. The accuracy found in that study was sufficient for gait classification, although the precision was less satisfactory. The performance of this algorithm might be improved by attaching the IMUs closer to the location of impact, i.e. the hoof of the horse, hence limiting the attenuation of the vibrations through the limb [14, 15].

During this study, two algorithms for automatic hoof- events detection based on the acceleration and angular velocity signals measured by hoof-mounted IMUs in walk and trot on a hard surface were developed. For gait classification, the needed accuracy and precision for hoof-event detection are not yet investigated. However, estimations of stance and swing durations in addition to knowledge about the timing of lateral and diagonal hoof placement are essential. We therefore aim for accuracies and precisions similar or better compared with the study of Bragança et al. (2017) [13].

## Materials and methods

At the start of this study, force plate and IMU data were visually examined. The IMU data showed distinctive peaks coinciding with the hoof-on and hoof-off times measured with the force plates, as previous described by Olsen et al. [9] and depicted in Fig 1. During the current study, we developed two algorithms to detect these distinctive peaks from the IMU data and applied a more advanced method to determine the hoof-on and hoof-off times from the force plate. These contact times of the force plate were used for the internal validation of the algorithms.

## Data collection

For the current study, we used data that was collected for a previous study [13]. Measurements were performed on seven Warmblood horses (*Equus ferus caballus*; for further details see S1 Appendix) in the Equine Clinic of Utrecht University at the Department Clinical Sciences.

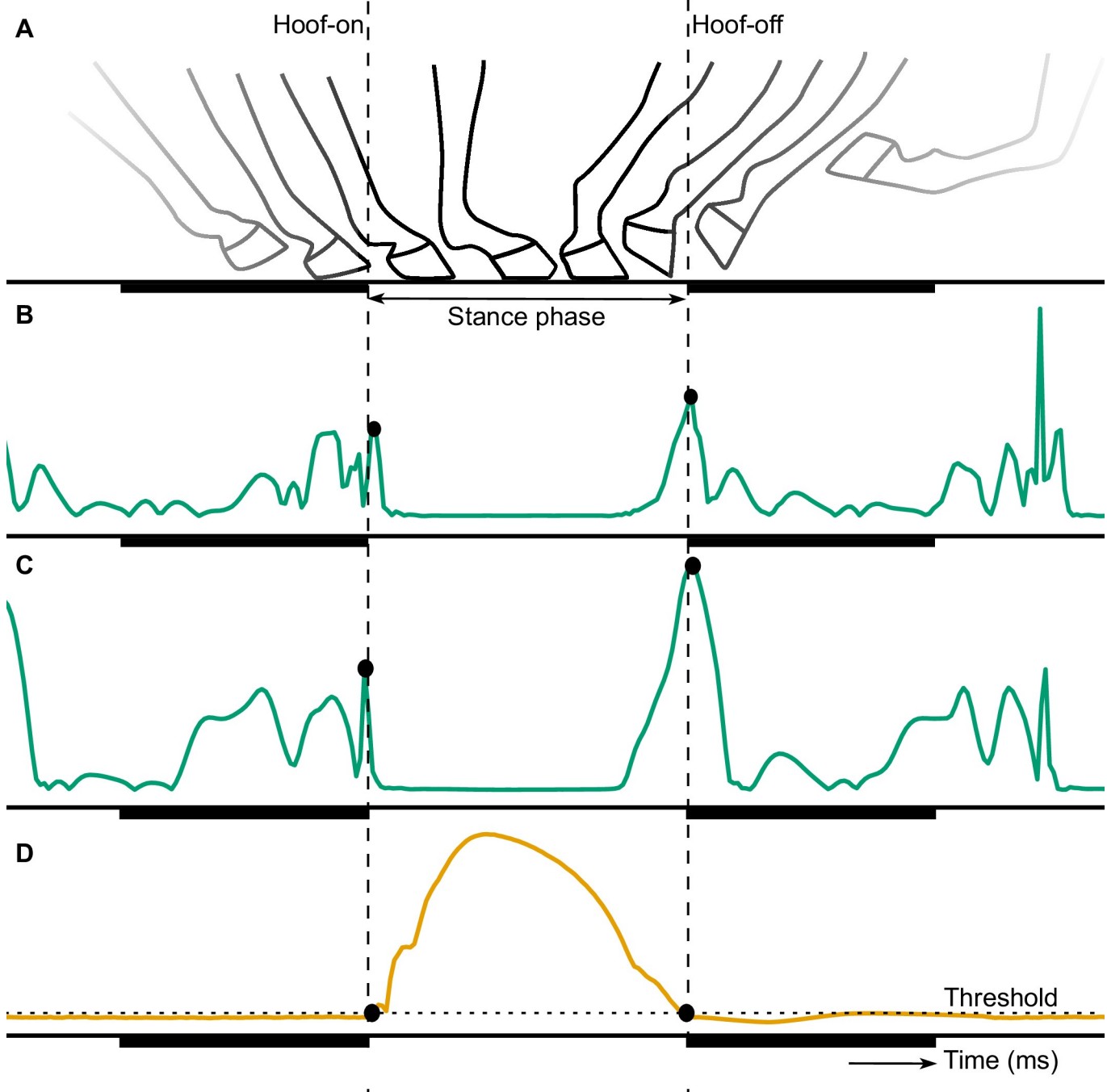

**Fig 1.** Generic illustration of the movement of the hoof (A), modified from Witte et al. [8], and the signals of the acceleration (B), angular velocity (C) and vertical force (D). The hoof-on and hoof-off events are depicted with the vertical dashed lines and the dots show the detected hoof-on and hoof-off events from the different signals. These hoof-events occur at the start and end of the stance phase, shown as the period not underlined by a dark beam. The horizontal dashed line in D shows the threshold used to detect the hoof-events from the force signal.

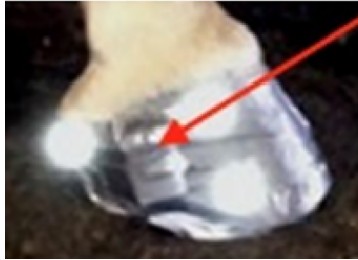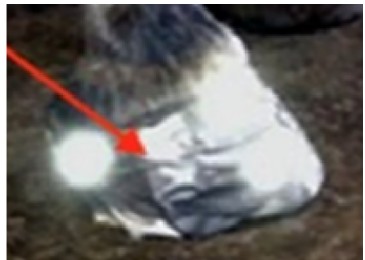

**Fig 2. Location of inertial measurement units (IMUs) on the hoof.** The location of the IMUs is indicated with red arrows, on the lateral quarter of the right front and hind hoof with reflective markers on both sides (lateral heel, lateral toe and lateral coronet) used for another study [13].

All horses were equipped with ProMove-mini wireless IMUs (Inertia-Technology B.V., Enschede, The Netherlands; for further details see S1 Appendix) which measured the acceleration, low-$g$ acceleration with a range of ±16 $g$ and high-$g$ acceleration with a range of ±400 $g$, and angular velocity, with a range of ±2000˚/s, and sampling frequency of 200 Hz. Two IMUs were attached to the lateral wall of the right front (RF) and hind (RH) hooves with double sided and normal tape as can be seen in Fig 2.

All horses were walked and trotted over a force plate (Z4852C, Kistler, Winterthur, Switzerland; for further details see S1 Appendix) to collect at least five valid force plate impacts for both front and hind hooves; each valid impact will be considered a trial in the further analysis. An impact was considered valid if two criteria were met: 1) only one entire hoof was placed on the force plate and 2) the horse was led in a straight line with a constant speed of 0.8 to 1.4 m/s for walk and 1.7 to 2.7 m/s for trot.

Three reflective markers of the OMC system (Qualisys AB, Motion Capture System, Göteborg, Sweden; for further details see S1 Appendix) were glued to lateral heel, lateral toe and lateral coronet of each hoof as can be seen in Fig 2. The collected OMC data were used in another study for break-over detection [16] but was needed for time synchronization in the current study.

The force plate and OMC system were time synchronized by a hardware connection ([13]; for further details see S1 Appendix). Time synchronization of the OMC system and the IMUs was accomplished by calculation of a cross-correlation between the angular velocity signal of the IMUs and the position signal of the reflective markers of the OMC system ([7, 13]; for further details see S1 Appendix).

The original horse measurements were performed in compliance with the Dutch Act on Animal Experimentation and approved by the local ethics committee of Utrecht University. All horses were present for teaching purposes and these measurements were not considered additional animal experiments within the Dutch law at that time. Therefore, no specific experiment number is available.

## Data analysis

**Force plate data.** The collected force plate data were preprocessed by Inertia Technology B.V.. The valid impacts were selected and cut into different trials; each trial consisted of at least one valid impact and sometimes two for consecutive impacts of the RF and RH hoof.

In Fig 1D, the vertical force signal of one valid impact can be seen. The dotted lines show the hoof-on and hoof-off time points, for the detection of which a threshold was used. This threshold value was calculated from the signal mean (x) and signal standard deviation (s) of the baseline, i.e. the period before the valid impact happened. To distinguish the impacts from

the baseline, the average of the force signal was calculated with a moving mean window with a length of 130 ms and the baseline was determined for average values below 100 N. For every trial, a threshold value (T) was determined by:

$$T = x + 2.58 \times s$$

The standard deviation was multiplied by 2.58 resulting in detection of the upper 0.5% of a normally distributed signal, to ensure that only high impacts are detected.

Hoof-on was determined as the first time point that the vertical force exceeded the threshold value. Hoof-off was determined as the first time point that the vertical force dropped below the threshold value.

**IMU data.** The collected IMU data were preprocessed by Inertia Technology B.V. and cut into different trials corresponding with the force plate trials. The collected IMU data consisted of two tri-axial acceleration signals, a low $g$ acceleration signal and a high $g$ acceleration signal, and one tri-axial angular velocity signal. The two acceleration signals were fused into one tri-axial acceleration signal that was used during the current study [7]. Further data analysis was performed in MATLAB (version R2017a, The MathWorks Inc., Natick, Massachusetts, USA).

The preprocessed tri-axial acceleration and angular velocity signals were further prepared for analysis in two steps: 1) offset drift was removed from the acceleration signal, and 2) the root of the sum of squares, Euclidean norm, was calculated of the tri-axial acceleration and angular velocity signals resulting in a one-directional acceleration and angular velocity signal. The Euclidean norm was used to reduce the calculation time in contrast to calculating a horse specific rotation matrix [11] and to cancel out artefacts due to wrong alignment of the sensor on the hoof. This will make the algorithms better applicable in field setting.

The preprocessed signal is shown in Fig 3A. To distinguish consecutive steps from each other, the stance phase and the swing phase of a limb were estimated by calculating the variance of the acceleration and angular velocity signals. The variance was calculated by applying a moving variance function over the two signals with window length of 130 ms. The variance of the angular velocity was higher than the variance of the acceleration signal and to accommodate for this we downscaled the variance of the angular velocity with a factor of twenty-five. The stance phase was determined when both signals had a variance below five, the remaining time points were allocated to the swing phase. The window length, downscale factor and variance threshold were kept the same for all horses and trials. These values were chosen to ensure that: 1) all time points of the swing phase were allocated to the estimated swing phase, and 2) every swing phase was succeeded by a stance phase. This procedure resulted in an estimated swing phase longer than the real swing phase to make sure that hoof-on and hoof-off events were included in the roughly estimated swing phase. The estimated swing phase is indicated by the box in Fig 3B.

Next, we determined the hoof-on and hoof-off from the acceleration and angular velocity signal separately by developing two algorithms.

The algorithm for the angular velocity signal assessed every swing phase separately. Peaks were detected, indicated by the dots in Fig 3B, and the mean peak height and mean peak prominence were calculated from these peaks. The peak prominence depicted how much the peak stands out due to its intrinsic height and location relative to the other peaks. Peaks were selected if the peak was higher than the mean peak height or the prominence was bigger than the mean prominence, or both. These selected peaks are indicated by the dots in Fig 3C. Hoof-off was determined as the time point corresponding with the peak closest to the start of the estimated swing phase. For hoof-on detection, only the detected peaks of the second half of the swing phase were assessed. Again, the mean peak height and mean peak prominence were

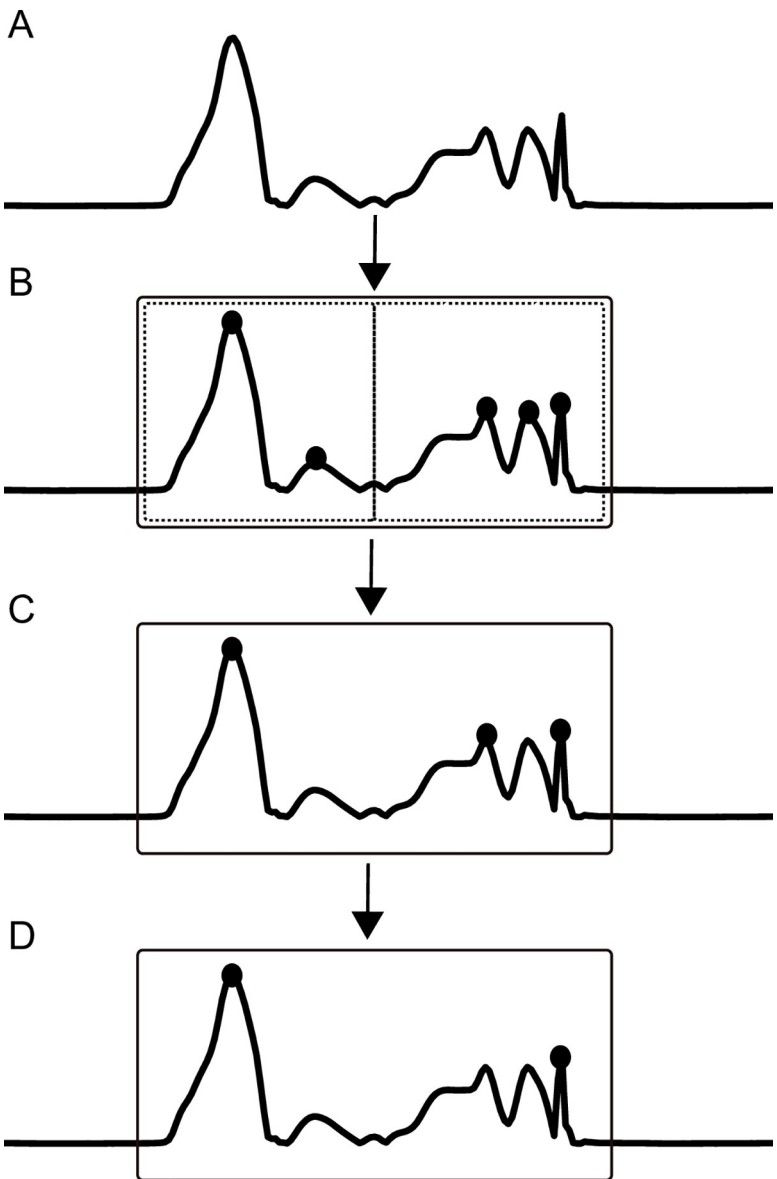

**Fig 3. Generic illustration of an IMU signal and the steps performed by both algorithms.** The preprocessed signal is indicated in A. The estimated swing phase is indicated with the box in B. For the angular velocity signal, the peaks within one estimated swing phase are detected (B). From these peaks, peaks were selected if the peak height or prominence was bigger than the mean peak height or prominence, or both (C). Thereafter, the peak closest to the start of the estimated swing phase was selected as the hoof-off time point and the peak closest to the end of the estimated swing phase was selected as the hoof-on time point (D). For the acceleration signal, these steps were performed for the first and second half of the estimated swing phase, indicated with the dotted boxes in B.

calculated. Peaks were selected if the peak was higher than the mean peak height or the prominence was bigger than the mean prominence, or both. Hoof-on was determined as the time point corresponding with the peak closest to the end of the estimated swing phase. The peaks selected as hoof-off and hoof-on are indicated with dots in Fig 3D.

The algorithm for the acceleration signal assessed the signal in a similar manner as described above. However, only the peaks detected in the first half of the swing phase were assessed for hoof-off detection and the peaks detected in the second half of the swing phase

were assessed for hoof-on. The first and second half of the swing phase are indicated by the dotted boxes in Fig 3B. After this step, peaks were detected, indicated by the dots in Fig 3B, and the mean peak height and mean peak prominence were calculated from these peaks. Peaks were selected if the peak was higher than the mean peak height or the prominence was bigger than the mean prominence, or both. These selected peaks are indicated by the dots in Fig 3C. Hoof-off was determined as the time point corresponding with the peak closest to the start of the estimated swing phase. Hoof-on was determined as the time point corresponding with the peak closest to the end of the estimated swing phase. The peaks selected as hoof-off and hoof-on are indicated with dots in Fig 3D.

**Stride parameter estimation.** With the determined hoof-on and hoof-off time points the following stride parameters were determined:

- Stance duration–time between hoof-on and hoof-off of the same hoof

- Hoof-on time difference–time difference between the hoof-on detection of both algorithms were assessed with the force plate hoof-on detection separately for a given hoof

- Hoof-off time difference–time difference between the hoof-off detection of both algorithms were assessed with the force plate hoof-off detection separately for a given hoof

## Performance evaluation

The normality of the stride parameters was visually checked by examining the QQ plot and histogram in R (version 1.1.414, RStudio Inc, Boston, Massachusetts, USA). Thereafter, the distribution of the hoof-on and hoof-off time differences was evaluated to interpret the results and the performance of both algorithms was evaluated by Bland Altman and linear mixed model analysis.

**Bland Altman.** The agreement between the acceleration algorithm and the force plate and the angular velocity algorithm and the force plate was evaluated for the stance duration. This evaluation compared two different methods to measure the stance duration and therefore a Bland Altman analysis was performed with the "BlandAltmanLeh" package [17].

The results of this analysis showed the mean difference in stance duration between the algorithms and the force plate and the standard deviation (SD) of these differences. These results were deemed better if closer to zero since this indicates a small and consistent difference between the algorithms and force plate, i.e. a good accuracy and precision. A positive mean indicates a shorter stance duration measured with the force plate and a negative mean indicates a longer stance duration measured with the force plate compared with the algorithms. In addition, the upper and lower confidence interval limits were used to calculate the width of the confidence interval. The width of the confidence interval was preferred to be small which means that the differences between the algorithms and the force plate were consistent.

**Linear mixed model analysis.** A linear mixed model analysis was performed to estimate the effect of horse, hoof, gait and trial on the performance of the algorithms for all stride parameters. This analysis was performed with the "lme4" package [18]. The independent variables of this analysis were the effect of hoof, gait, number of analyzed trials and interaction term between hoof and gait. The model is described by:

$$Y_{ijkl} \sim \mu + \text{hoof}_i + \text{gait}_j + \text{trial}_k + (\text{hoof x gait})_{ij} + (1|\text{horse}) + \varepsilon_{ijkl}$$

where $Y_{ijkl}$ is the predicted value of the *ijkl*-th record, $\mu$ is the overall mean, $\text{hoof}_i$ is the effect of hoof (*i* can be RF or RH hoof), $\text{gait}_j$ is the effect of gait (*j* can be walk or trot), $\text{trial}_k$ is the effect of trial (*k* can be 1 until 9 depending on the number of trials collected for a horse), (hoof

x gait)$_{ij}$ is the effect of the interaction between hoof$_i$ and gait$_j$ and $\varepsilon_{ijkl}$ is the residual error term associated with the *ijkl*-th record. A random intercept for every horse was included in the model.

Model reduction was applied with the Akaike's information criterion and the model with the lowest Akaike's information criterion values was selected according to Occam's Razor principle. The residuals of each selected model were visually inspected and checked for any deviations of normality and homoscedasticity. The predicted value of the stance duration and the time difference between the algorithms and the force plate (Y) were calculated for every combination of hoof$_i$ and gait$_j$ ("emmeans" package [19]). In addition, the lower and upper limits were calculated of the 95% confidence interval ("MASS" package [20]).

The performance of these algorithms was evaluated based on the predicted values and width of the confidence intervals. For the stance durations, the predicted values of both algorithms were deemed better if closer to the predicted value of the force plate and the width of the confidence interval was preferred to be small, which indicates a good precision. For the hoof-on and hoof-off time differences between the algorithms and the force plate, the predicted value was deemed better if closer to zero since this indicates a small difference between the algorithms and force plate, i.e. a good accuracy. A positive predicted value indicates a delayed detection by the algorithms and a negative predicted value indicates a too early detection by the algorithms compared with the force plate measurement. The width of the 95% confidence interval was preferred to be small, which means that the differences between the algorithms and the force plate were consistent, i.e. a good precision. Schematic representations of these predicted values were used to evaluate the accuracy and precision of the algorithms.

## Results

A total of 147 trials were analyzed: 75 trials of the right front (RF) hoof (36 in walk and 39 in trot) and 72 trials of the right hind (RH) hoof (34 in walk and 38 in trot). In Table 1 an overview is given of the number of the analyzed trials and hoof and gait characteristics. Preprocessed data of one measurement in trot can be seen in S1 Fig. Stance durations were calculated and can be found in S1 Table. The stride parameters were normally distributed.

The distributions of the time differences for hoof-on detection are illustrated in Fig 4A for the acceleration algorithm and in Fig 4B for the angular velocity algorithm. The distribution in both figures show higher values for the RH hoof in walk and lower values for the RF hoof in trot. The distribution in Fig 4A has a mean around 7 ms in contrast to the distribution in Fig 4B which has a mean of 16.5 ms. Furthermore, in Fig 4A there are no outliers in contrast to Fig 4B in which the distribution has outliers around -75 and 50 ms.

The distributions of the time differences for hoof-off detection are illustrated in Fig 4C for the acceleration algorithm and in Fig 4D for the angular velocity algorithm. The distribution in Fig 4C has a lower mean, around 0.78 ms, compared to the distribution in Fig 4D which has a mean of 3.2 ms. In Fig 4C, the distribution has outliers around -57.5, 55 and 150 ms in

**Table 1. Number of analyzed trials collected per horse, gait and hoof.**

| horse ID | | 1 | 2 | 3 | 4 | 5 | 6 | 7 | total |
|---|---|---|---|---|---|---|---|---|---|
| Walk | RF | 5 | 5 | 5 | 5 | 5 | 6 | 5 | 36 |
| | RH | 5 | 5 | 5 | 4 | 5 | 5 | 5 | 34 |
| Trot | RF | 5 | 5 | 6 | 5 | 8 | 5 | 5 | 39 |
| | RH | 7 | 5 | 5 | 5 | 5 | 6 | 5 | 38 |
| | | 22 | 20 | 21 | 19 | 23 | 22 | 20 | 147 |

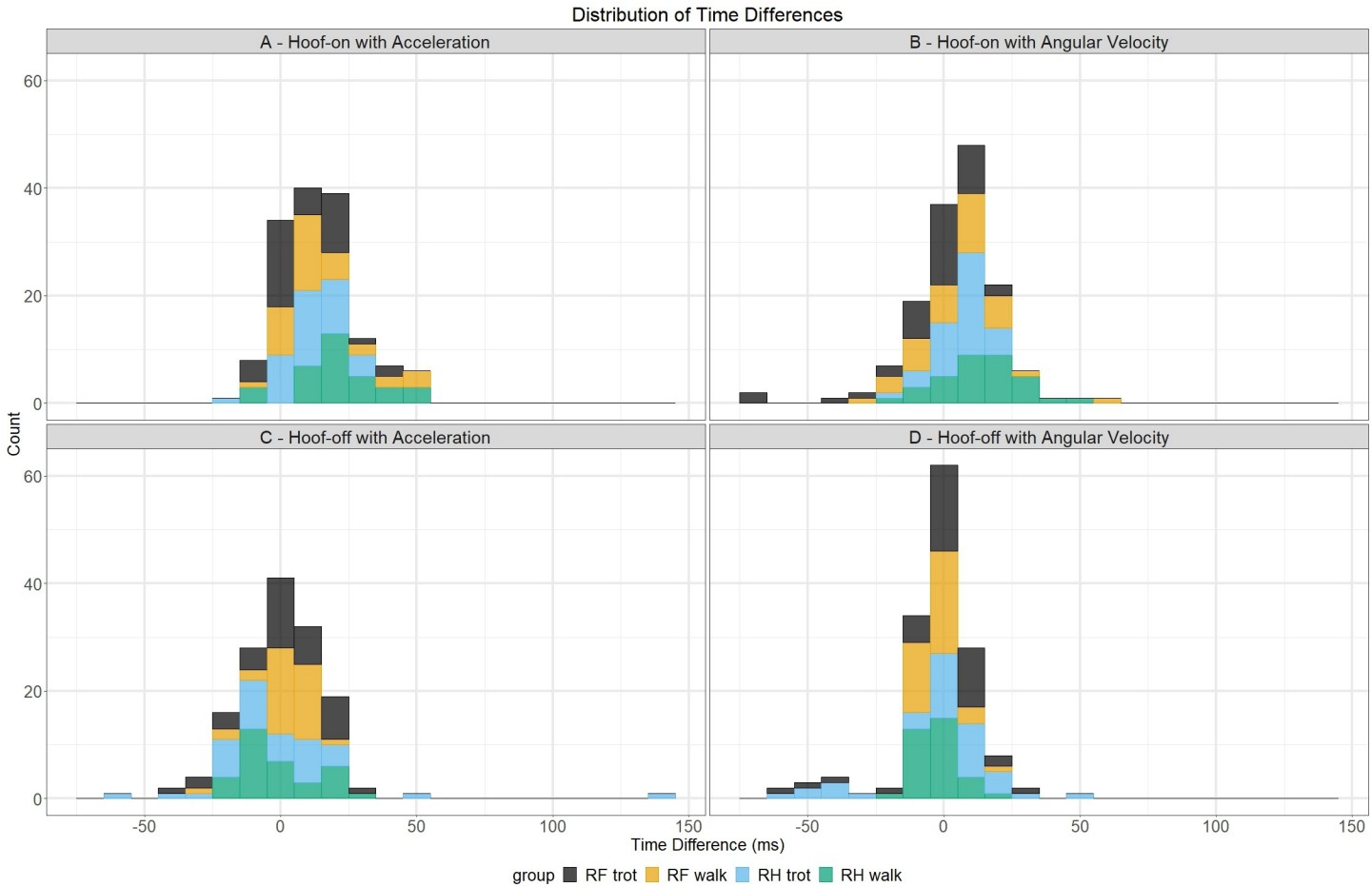

**Fig 4. Distributions of time differences between both algorithms and the force plate for hoof-on and hoof-off detection.** Time differences for hoof-on detection are depicted in the upper row and time differences for hoof-off are depicted in the bottom row. The different hoof/gait combinations are depicted with their own color.

contrast to the distribution in Fig 4D which has outliers around -50 and 50 ms. For both models, no clear distinction could be made between the different hoof/gait combinations.

## Bland Altman analysis

The results in Table 2 show that the mean difference and SD were closer to zero for the angular velocity algorithm, except for the SD of the RF hoof in trot, which was higher compared to the acceleration algorithm. Also, the confidence intervals were smaller for the angular velocity algorithm, except for the RF hoof in trot which is caused by a higher SD. These results indicate that the agreement with the force plate was, in general, better for the angular velocity algorithm for the stance duration. Furthermore, the mean difference was negative for all groups, except for the RF hoof in trot, which means that shorter stance durations were measured with both algorithms compared to the force plate.

## Linear mixed model analysis

The residuals of all selected linear mixed models were normally distributed and did not show homoscedasticity.

**Table 2. Bland Altman results for stance duration.**

| | | | mean (ms) | SD (ms) | lower CI (ms) | upper CI (ms) |
|---|---|---|---|---|---|---|
| | | **Stance duration** | | | | |
| acceleration | walk | RF | -2.67 | 3.76 | -10.05 | 4.71 |
| | | RH | -4.18 | 3.52 | -11.08 | 2.72 |
| | trot | RF | -1.64 | 3.84 | -9.17 | 5.89 |
| | | RH | -2.39 | 6.18 | -14.52 | 9.73 |
| angular velocity | walk | RF | -1.33 | 3.20 | -7.60 | 4.94 |
| | | RH | -2.88 | 2.86 | -8.48 | 2.72 |
| | trot | RF | 0.74 | 4.98 | -9.01 | 10.50 |
| | | RH | -1.66 | 4.52 | -10.52 | 7.20 |

The mean differences in stance duration between the algorithms and force plate in milliseconds (ms) and the standard deviation (SD) of this mean difference in ms are deemed better if closer to zero. The 95% confidence interval was preferred to be small.

**Hoof-on detection.** The results presented in Table 3 are the models with the lowest AIC values. The predicted values of the time differences between the acceleration algorithm and the force plate (model 1) were best explained with hoof, gait, trial and interaction term as fixed effect and horse as random effect. For this model, the predicted values and lower and upper confidence interval limits were averaged over the number of analyzed trials. For the time differences between the angular velocity and the force plate (model 2), hoof and gait were needed as fixed effects and horse as random effect to explain the data best.

The results in Table 3 show that the predicted values of the time differences were smaller for the angular velocity algorithm (model 2) compared with the acceleration algorithm (model 1). All predicted values were positive which indicates a delayed detection by both algorithms compared with the force plate. Also, the confidence intervals were smaller for the angular velocity algorithm.

In Fig 5A, the predicted values and their confidence intervals of model 1 are shown for every hoof/gait combination because this model needs an interaction term to explain the data. In Fig 5B, the predicted values and their confidence intervals of model 2 are shown for walk

**Table 3. Linear mixed model results for the time differences in hoof-on and hoof-off detection.**

| | | | predicted value (ms) | lower CI (ms) | upper CI (ms) |
|---|---|---|---|---|---|
| | | **Hoof-on time differences** | | | |
| Model 1: acceleration | walk | RF | 17.93 | 9.33 | 26.52 |
| | | RH | 23.96 | 15.35 | 32.57 |
| | trot | RF | 13.77 | 5.20 | 22.34 |
| | | RH | 14.84 | 6.27 | 23.41 |
| Model 2: angular velocity | walk | | 11.06 | 4.13 | 17.99 |
| | trot | | 3.55 | -3.35 | 10.45 |
| | | RF | 2.39 | -4.52 | 9.30 |
| | | RH | 12.22 | 5.29 | 19.14 |
| | | **Hoof-off time differences** | | | |
| | | | predicted value (ms) | lower CI (ms) | upper CI (ms) |
| Model 3: acceleration | | | 3.20 | 0.05 | 6.34 |
| Model 4: angular velocity | | | 0.75 | -3.83 | 5.32 |

The predicted values of the time difference between both algorithms relative to the force plate are determined in milliseconds (ms) and are deemed better if closer to zero. The 95% confidence interval was preferred to be small.

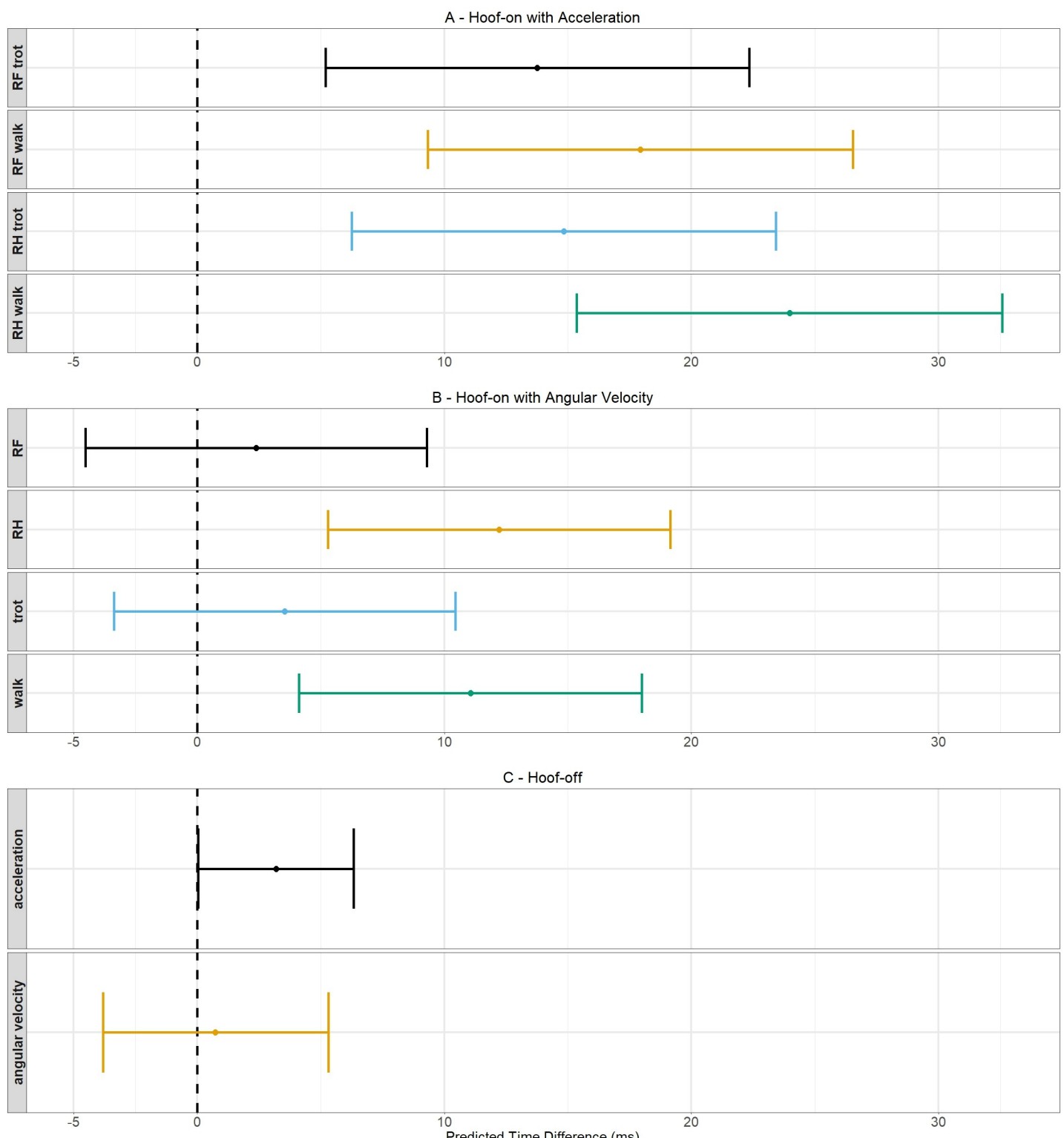

**Fig 5. Schematic representation of the predicted values of the time differences and their 95% confidence intervals.** The dots indicate the predicted value for a certain hoof/gait combination and the 95% confidence intervals are shown by the whiskers. The dashed line indicates a predicted time difference of 0 ms.

**Table 4. Linear mixed model results for stance duration.**

| Stance duration | | | predicted value (ms) | lower CI (ms) | upper CI (ms) |
|---|---|---|---|---|---|
| acceleration | walk | RF | 779.71 | 759.13 | 800.28 |
| | | RH | 777.97 | 757.18 | 798.76 |
| | trot | RF | 337.06 | 316.73 | 357.39 |
| | | RH | 302.57 | 282.17 | 322.97 |
| angular velocity | walk | RF | 786.53 | 763.98 | 809.08 |
| | | RH | 784.63 | 761.94 | 807.32 |
| | trot | RF | 349.01 | 326.61 | 371.41 |
| | | RH | 306.32 | 283.88 | 328.76 |
| force plate | walk | RF | 793.34 | 771.89 | 814.80 |
| | | RH | 799.19 | 777.60 | 820.78 |
| | trot | RF | 345.32 | 324.02 | 366.61 |
| | | RH | 314.96 | 293.62 | 336.30 |

The predicted value for the stance duration is determined in milliseconds (ms) for both algorithms and the force plate. The 95% confidence interval was preferred to be small.

versus trot and RF versus RH hoof because this model did not need an interaction term to explain the data. The predicted values are located closer to zero for model 2 and their confidence intervals are smaller.

These results indicate that the agreement with the force plate was, in general, better for the angular velocity algorithm with an accuracy between 2.39 and 12.22 ms depending on the gait and hoof and a precision of around 13.83 ms for the hoof-on detection.

**Hoof-off detection.** The predicted values of the time differences between the acceleration algorithm and the force plate (model 3) were best explained with an empty model with no random effect. For the time differences between the angular velocity algorithm and the force plate (model 4), an empty model with random effect for horse was needed to explain the data best.

The results in Table 3 show that the predicted value was smaller for the angular velocity algorithm (model 4) compared with the acceleration algorithm (model 3). Both predicted values are positive which indicates a delayed detection by both algorithms compared with the force plate. The confidence interval was smaller for the acceleration algorithm. In Fig 5C, a schematic representation of these findings is shown.

These results indicate that the agreement with the force plate was better for the acceleration algorithm with an accuracy of 3.20 ms and precision of 6.39 ms for hoof-off detection.

**Stance duration.** For all three models, the predicted values for the stance duration were best explained when hoof, gait and interaction term were included as fixed effect and horse as random effect in the model. The results in Table 4 show that the predicted values of both algorithms are smaller compared with the force plate, except for the RF hoof in trot of the angular velocity algorithm. The predicted values determined with the acceleration algorithm are the lowest. Also, the width of the confidence intervals of both algorithms were smaller than the intervals of the force plate, except for the RF hoof in trot of the angular velocity algorithm. These results agree with the results of the Bland Altman analysis.

## Discussion

Two algorithms are presented to automatically detect hoof-events from the acceleration and angular velocity signals measured with hoof-mounted IMUs in horses walking and trotting on

hard ground. Results of internal validation with the force plate showed that, for the hoof-on detection, the angular velocity algorithm was the most accurate with an accuracy between 2.39 and 12.22 ms and a precision of around 13.80 ms, depending on gait and hoof. For hoof-off detection, the acceleration algorithm was the most accurate with an accuracy of 3.20 and precision of 6.39 ms, independent of gait and hoof.

From the results we can conclude that hoof-on is better detected by the angular velocity algorithm which might be explained by the fact that the hoof will slide forward after vertical impact on a hard surface. The forward slide results in a silent angular velocity signal while the acceleration is not silent. Also, a difference in accuracy between the RF hoof (2.39 ms) and RH hoof (12.22 ms) was found which can be explained by the fact that horses place their front and hind hooves differently on the ground. In previous studies, also different landing and braking characteristics are found for hind, front, leading and trailing limbs [21–23]. Furthermore, the front hooves bounce more at impact in contrast to the hind hooves, which slide more at impact [24]. For the hoof-off detection, the acceleration algorithm performed better which might be explained by the gradual hoof rotation prior to hoof-off. This gradual rotation results in an increase in the angular velocity signal while the acceleration signal is more strongly increased at the actual hoof-off moment. These phenomena could be different and variable on surfaces with other properties. Less firm surface material, such as sand would allow penetration of the hoof into the substrate. If the surface offers shear resistance the hoof would slide less forward [25]. This could alter the appearance of the angular velocity versus the acceleration signal. Thomason and Peterson (2008) described a more evident forward push when the surface is smooth and firm [26]. Since these algorithms are only tested on data measured on a hard surface, more extensive studies should be performed in to validate it for other surfaces.

In a previous study by Bragança et al. (2017), accuracy and precision for hoof-on were slightly better for the RH hoof and similar for the RF hoof. For hoof-off detection, the accuracy and precision found in this study were better [13]. It was expected to find a better algorithm performance during this study due to the use of hoof-mounted IMUs. However, this expectation was only met for the hind hoof and not for the front hoof.

In another study, algorithms were developed to detect gait events from OMC data. Validation with the force plate showed an accuracy between -13.6 and 21.5 ms and a precision between 5.8 and 32.9 ms, depending on limb and gait [27], which is almost similar to the IMU algorithms of Bragança et al. (2017) [13]. So, no clear distinction in performance could be made between algorithms developed for OMC data and IMU data. However, both studies validated these algorithms against the force plate.

In previous studies, reservations about the use of the force plate as gold standard for lameness detection are described [28, 29]. They reported that some parameters measured with the force plate should be considered less reliable than others [28, 29]. Furthermore, detection of stance duration was performed by using a threshold for the force plate signal. The use of a threshold value is arbitrary; therefore a trial specific threshold was calculated in the study of Clayton et al. (1999) [30] to eliminate the horse-specific aspects, such as walking speed and weight of the horse, and the effect of noise on the signal. Stance durations determined with the IMUs were shorter than the durations determined with the force plate which is probably caused by the threshold level used for the force plate signal since stance and swing phases are estimated from the IMU signals by calculating the variance of the signals. Also, other studies described differences in stance duration according to the threshold levels used for the force plate signal [4]. The reason that we chose to use the force plate as gold standard is that this system is used in most research facilities.

The OMC system used guarantees a relative precision of 1.9 mm [7] measuring the kinematics of the hoof and introduces different definitions of hoof-on and hoof-off such as toe-on,

heel-on, toe-off and heel-off timings. Therefore, the OMC system might be a more appropriate technique to study the hoof movement and break-over phase more in detail. During this study, the break-over phase was also included in the analysis but is described elsewhere [16].

The needed accuracy and precision for gait classification are not yet determined. Stance duration are measured in different gaits at different speeds and the shortest stance duration reported was 103 ms in pace [31]. Therefore, an algorithm with an accuracy and precision smaller than 100 ms might be sufficient to detected foot-fall pattern and thus gait classification. For lameness detection however, a more accurate and precise algorithm is needed since the stance duration increases with 1% in both the affected and contralateral limbs for mild lameness. [32, 33].

## Conclusion

Two algorithms are presented to automatically detect hoof-on and hoof-off from acceleration and angular velocity data measured with hoof-mounted IMUs in walk and trot on a hard surface. Internal validation against the force plate was performed. The results showed that for the hoof-on detection, the angular velocity algorithm was the most accurate with an accuracy between 2.39 and 12.22 ms and a precision of around 13.80 ms, depending on gait and hoof. For hoof-off detection, the acceleration algorithm was the most accurate with an accuracy of 3.20 ms and precision of 6.39 ms, independent of gait and hoof. These algorithms seem promising for gait classification, although a more extensive validation process should be performed. Also, the applicability of these algorithms should be investigated under different circumstances, such as different ground surfaces, gaits, speed and different hoof trimming conditions.

## Supporting information

**S1 Appendix. Additional information.** Document with additional information about the population, data collection and synchronization of all the measurement systems.
(DOCX)

**S2 Appendix. Dataset.** Excel file with hoof-on, hoof-off and break-over data.
(XLSX)

**S1 Table. Table with stance durations.** Table with stance durations as detected with the algorithms for acceleration, angular velocity and force plate.
(DOCX)

**S1 Fig.** Preprocessed signals of the vertical force (A), the acceleration (B) and angular velocity (C) signals of the IMU for one hoof from one measurement in trot. The hoof-on events are depicted with upward-pointing triangle markers and hoof-off events are depicted with downward-pointing triangle markers.
(TIF)

## Acknowledgments

We would like to thank W. Back, M. Marin-Perianu and P.R. van Weeren for making this study possible. A special thanks to W. Back and P.R. van Weeren for the feedback on preliminary results of the current study and J. van den Broek for statistical guidance.

## Author Contributions

**Conceptualization:** F. M. Serra Bragança.

**Data curation:** M. Tijssen, F. M. Serra Bragança.

**Formal analysis:** M. Tijssen, S. Bosch, F. M. Serra Bragança.

**Investigation:** S. Bosch, J. P. Voskamp, F. M. Serra Bragança.

**Methodology:** M. Tijssen, M. Nielen.

**Project administration:** J. P. Voskamp.

**Resources:** F. M. Serra Bragança.

**Software:** M. Tijssen, F. M. Serra Bragança.

**Supervision:** M. Nielen, F. M. Serra Bragança.

**Validation:** M. Tijssen, M. Nielen.

**Visualization:** M. Tijssen, M. Nielen.

**Writing – original draft:** M. Tijssen.

**Writing – review & editing:** M. Tijssen, E. Hernlund, M. Rhodin, S. Bosch, J. P. Voskamp, M. Nielen, F. M. Serra Bragança.

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
