## [Decision Letter · Decision Letter 0]

25 Mar 2020

PONE-D-19-27743

Automatic hoof-on and -off detection in horses using hoof-mounted inertial measurement unit sensors

PLOS ONE

Dear Drs. Tijssen,

Thank you for submitting your manuscript to PLOS ONE. After careful consideration, we feel that it has merit but does not fully meet PLOS ONE’s publication criteria as it currently stands. Therefore, we invite you to submit a revised version of the manuscript that addresses the points raised during the review process.

We would appreciate receiving your revised manuscript by May 09 2020 11:59PM. To enhance the reproducibility of your results, we recommend that if applicable you deposit your laboratory protocols in protocols.io, where a protocol can be assigned its own identifier (DOI) such that it can be cited independently in the future. For instructions see: http://journals.plos.org/plosone/s/submission-guidelines#loc-laboratory-protocols

We look forward to receiving your revised manuscript.

Kind regards,

Chris Rogers

Academic Editor

PLOS ONE

Additional Editor Comments (if provided):

Thank you for your patience with this submission. I apologize for the delay, but it has proven difficult to obtain reviewers for this manuscript, partly because your group has so many collaborators. The reviewers were generally complementary of the manuscript and the comments are really of an editorial nature and should provide greater clarity for readers not embedded in the field of equine biomechanics. I doubt it will take long to make the necessary emendations. i will have a look on the editor manager software and see if i can extend the due date for the revision of the companion manuscript to this so you can ensure complementary changes etc.

Journal Requirements:

"I have read the journal's policy and the authors of this manuscript have the following competing interests: M. Marin-Perianu founded Inertia-Technology B.V., which sells the inertial sensor system (Promove-mini) that is used as the basis of the EquiMoves® system, which is evaluated in this study. S. Bosch is an employee of Inertia-Technology B.V.. Rosmark Consultancy and Utrecht University are partners in the EquiMoves® corporation."

Please know it is PLOS ONE policy for corresponding authors to declare, on behalf of all authors, all potential competing interests for the purposes of transparency. PLOS defines a competing interest as anything that interferes with, or could reasonably be perceived as interfering with, the full and objective presentation, peer review, editorial decision-making, or publication of research or non-research articles submitted to one of the journals. Competing interests can be financial or non-financial, professional, or personal. Competing interests can arise in relationship to an organization or another person. Please follow this link to our website for more details on competing interests: ht3tp://journals.plos.org/plosone/s/competing-interests

"W. Back, 13448, STW Valorisation Grant, https://www.tudelft.nl/, The funders had no role in study design, data collection and analysis, decision to publish, or preparation of the manuscript.

M. Rhodin, H-17-47-303, Swedish-Norwegian Foundation for Equine Research, https://hastforskning.se/, The funders had no role in study design, data collection and analysis, decision to publish, or preparation of the manuscript."

We note that one or more of the authors are employed by a commercial company: Inertia Technology B.V. and Rosmark Consultancy.

Reviewers' comments:

Reviewer's Responses to Questions

**Comments to the Author**

1. Is the manuscript technically sound, and do the data support the conclusions?

Reviewer #1: Yes

Reviewer #2: No

2. Has the statistical analysis been performed appropriately and rigorously? 

Reviewer #1: I Don't Know

Reviewer #2: Yes

3. Have the authors made all data underlying the findings in their manuscript fully available?

Reviewer #1: Yes

Reviewer #2: Yes

4. Is the manuscript presented in an intelligible fashion and written in standard English?

Reviewer #1: Yes

Reviewer #2: Yes

5. Review Comments to the Author

Reviewer #1: The sampling rate of the IMU used is only 200 samples per second, so time resolution is already greater than 5 ms so insufficient, especially for faster gaits.

Using the force plate as a measurement standard does not have a strong justification.

The force plate measures force while the IMU measures motion. Therefore, it is not clear what the time relationship is between the resolved acceleration vector and the force plate loading during hoof impact and hoof rollover.

No mention is made of the response time lag of the force plate signal.

If the aim is to compare timing events measured with an IMU between limbs, or on the same limb under different conditions, it would appear to be circuitous to use a force measurement as a reference.

Miniature high sampling rate IMU systems and recorders have variable integrated circuit clock speeds. It is important when trying to achieve timing accuracies to less than 5 ms that the recording systems are all calibrated. It is not clear how the force plate and IMU time records were synchronised or aligned.

Until the required level of timing accuracy from the IMU can be demonstrated for a single horse there would appear to be little value in evaluating the system with a larger number of horses.

Reviewer #2: The study presented a method to detect timings of hoof-on and off in horses using IMU sensors attached to the hooves. I have the following concerns regarding the presentation of the results and discussions associated with them. These are in no particular order.

1) The authors claimed that the angular velocity algorithm was more useful for accurate detection of the hoof-on timing whereas the acceleration algorithm was better for the detection of the hoof-off timing. I was quite surprised with this result because I expected that the acceleration signal was more useful for the hoof-on detection because the sudden change in the velocity (i.e. large magnitude of acceleration) of the hoof occurs due to the collision, but less useful for the hoof-off detection because the hoof just leaves the ground and no collision takes place. The authors tried to explain the reasons behind by explaining the behaviors of the hoofs in the hoof-on and off events in the Discussion section, but no actual data about the movement of the hooves were provided. I think it is necessary to provide the actual movement data of the hooves during locomotion measured using the motion capture system to warrant the provided explanations.

2) Figure 1 presented a schematic representation of the movements of the hoof, but not actual data measured by the IMU, force plate and motion capture system were provided. Why not provide actual data? Actually measured acceleration and angular velocity profiles from the IMUs, displacement profiles from the motion capture system and ground reaction force profiles from the force plate of both fore and hindlimb hooves from at least one representative trial should be provided to provide readers a better picture of what’s really happening to the hooves during locomotion.

3) Means and standard deviations of some basic spatiotemporal parameters of analyzed walking and trotting trials such as stride time, stride length, and stance time should be provided. Such information is necessary to determine the necessary sampling frequency and needed accuracy of the system.

4) Time scale should be provided in figures 1 and 3.

5) When using an IMU for motion analyses, what is always a problem is a drift. How did the authors cope with this problem in this study?

6) Why did the authors conduct both Bland Altman and linear mixed model analysis? I think the former is just sufficient for the evaluation of the algorithm because it is more direct and the two analyses essentially draw the same conclusion here.

6. PLOS authors have the option to publish the peer review history of their article (what does this mean?). If published, this will include your full peer review and any attached files.

Reviewer #1: No

Reviewer #2: No

---

## [Author Response · Author response to Decision Letter 0]

1 May 2020

Reviewer #1:

1. The sampling rate of the IMU used is only 200 samples per second, so time resolution is already greater than 5 ms so insufficient, especially for faster gaits.

Indeed, the sampling rate of the IMUs was set to 200 Hz in this study. In other studies, IMUs were set to a sampling frequency of 500 Hz to be able to measure faster gaits like pace and tölt as described in the abstracts of Gunnarsson et al. 2018 and Serra Braganca et al. 2018 (2, 3). The sampling frequency of the IMUs can be further increased to 1000 Hz if needed.

2. Using the force plate as a measurement standard does not have a strong justification. The force plate measures force while the IMU measures motion. Therefore, it is not clear what the time relationship is between the resolved acceleration vector and the force plate loading during hoof impact and hoof rollover.

Thank you for your comment. We agree that the force plate and IMUs measure another quantity which makes it difficult to compare the output of both systems directly. In this paper, we compare the timing of hoof-on and hoof-off. We detect these timings based on algorithms adapted to the specific signal and signal behavior, so whether the signal predicts vertical force, acceleration or angular velocity is not of importance to us. For this paper, we chose to use the force plate as measurement standard because this system is used in most research facilities and is still considered the gold standard for kinetic measurements (1, 4). For the break-over, the rolling motion of the hoof, the quantity of the measurement system becomes more important. Therefore, we used the OMC system as additional reference measurement for break-over detection onset as described in companion paper (5).

3. No mention is made of the response time lag of the force plate signal. Miniature high sampling rate IMU systems and recorders have variable integrated circuit clock speeds. It is important when trying to achieve timing accuracies to less than 5 ms that the recording systems are all calibrated. It is not clear how the force plate and IMU time records were synchronized or aligned.

We agree that the time synchronization between the different measurement systems was not clearly described. We have added a section about time synchronization in the supplementary file S1 and elaborated a little bit more on this in main method section in line 115-118. The response time lag of the force plate was 5.5 ms (for more details: page 134 of the QTM manual (6)) and is removed during preprocessing of the data.

4. If the aim is to compare timing events measured with an IMU between limbs, or on the same limb under different conditions, it would appear to be circuitous to use a force measurement as a reference.

Thank you for your feedback. The aim of this article was not to compare timing events under different conditions or on the same limb under different conditions. The aim of this paper was to develop an algorithms to automatically detect the hoof-on and -off events from the acceleration and angular velocity signal measured with IMUs. To validate these algorithms, hoof-on and hoof-off timings measured with the force plate were used since this is the gold standard for kinetic studies and is widely used in the clinics.

5. Until the required level of timing accuracy from the IMU can be demonstrated for a single horse there would appear to be little value in evaluating the system with a larger number of horses.

Thank you for your comment. We agree that the required level of accuracy is not clear yet. The level of accuracy needed depends on its purpose, e.g. for gait classification less accuracy is needed compared to lameness detection. Furthermore, also the type of surface will be of importance as described in the discussion. In this study, we made a start by developing an algorithm for automated hoof-on and hoof-off detection. We used data of two legs per horse (one front and one hind) and we analyzed consecutive steps to increase the number of measurements although the number of horses was small. The next step will be to determine the needed accuracy and precision for different purposes and different surface types, and to perform another study with a larger number of horses to determine the required level of accuracy.

Reviewer #2:

The study presented a method to detect timings of hoof-on and off in horses using IMU sensors attached to the hooves. I have the following concerns regarding the presentation of the results and discussions associated with them. These are in no particular order.

1) The authors claimed that the angular velocity algorithm was more useful for accurate detection of the hoof-on timing whereas the acceleration algorithm was better for the detection of the hoof-off timing. I was quite surprised with this result because I expected that the acceleration signal was more useful for the hoof-on detection because the sudden change in the velocity (i.e. large magnitude of acceleration) of the hoof occurs due to the collision, but less useful for the hoof-off detection because the hoof just leaves the ground and no collision takes place. The authors tried to explain the reasons behind by explaining the behaviors of the hoofs in the hoof-on and off events in the Discussion section, but no actual data about the movement of the hooves were provided. I think it is necessary to provide the actual movement data of the hooves during locomotion measured using the motion capture system to warrant the provided explanations.

Thank you for your comment. We were also quite surprised, at first, with the finding that the angular velocity was more accurate for the hoof-on detection and the acceleration for the hoof-off detection. After some consideration of the sliding and braking characteristics of the hoof, described by Hernlund et al. 2010 (7), we really think that the explanation as given in the discussion section, is valid for this finding. We understand that actual movement data of the hoof would be helpful to visualize the movement of the hoof and might help to understand this explanation. However, adding position and angular data of the hoof requires adding the OMC data to this paper. After careful consideration, we think that this is beyond the scope of this paper for multiple reasons. Firstly, the OMC data show more complicated data regarding the kinematics of the hoof, i.e. position of the markers and angles of the markers relative to other markers. Adding this information would make the paper more technical and harder to read. Secondly, this type of data is relevant when studying angles or relative position and we therefore included the OMC data in the companion paper (5) where we described the break-over onset detection. Furthermore, adding OMC data would also introduce the discussion of how to define the hoof-on moment. Namely, hoof-on can be determined as the toe-on or heel-on time point and we did prefer to stay away from this discussion. However, we understand the need to provide readers with actual data and therefore we have added a figure with data from the force plate, OMC system and IMU to the supplementary file.

2) Figure 1 presented a schematic representation of the movements of the hoof, but not actual data measured by the IMU, force plate and motion capture system were provided. Why not provide actual data? Actually measured acceleration and angular velocity profiles from the IMUs, displacement profiles from the motion capture system and ground reaction force profiles from the force plate of both fore and hindlimb hooves from at least one representative trial should be provided to provide readers a better picture of what’s really happening to the hooves during locomotion.

3) Time scale should be provided in figures 1 and 3.

We understand your concern and want to explain that figure 1 and 3 are generic illustrations of the IMU and force plate signals. These illustrations are based on actual data that is manipulated to illustrate the reader for which features/characteristics the algorithm is looking in the signal. We think that adding a time scale does not provide extra information to these illustrations since the steps of a horse are repetitive behavior that is not bound to one specific time interval. Actual data of one measurement is added to the supplementary file S1. Furthermore, we have changed the captions of figure 1 and 3 (line 86 and 163) to avoid misinterpretation.

4) Means and standard deviations of some basic spatiotemporal parameters of analyzed walking and trotting trials such as stride time, stride length, and stance time should be provided. Such information is necessary to determine the necessary sampling frequency and needed accuracy of the system.

Good suggestion, we have added a table with the mean and standard deviation values of the stance durations detected by the acceleration, angular velocity and force plate to the supplementary file S1. During this study, stride lengths were not analyzed, although this is possible the method to do this not yet validated and beyond the scope of this current study.

5) When using an IMU for motion analyses, what is always a problem is a drift. How did the authors cope with this problem in this study?

Indeed, you always have to be careful about drift. In this paper, we work with the acceleration and angular velocity data and not the integrated data to avoid drift. Furthermore, we work with one-directional data by calculating the Euclidean norm to avoid orientational issues as well.

6) Why did the authors conduct both Bland Altman and linear mixed model analysis? I think the former is just sufficient for the evaluation of the algorithm because it is more direct, and the two analyses essentially draw the same conclusion here.

Thank you for your question. Indeed, a Bland Altman test is more direct and easier to understand analysis method compared with a linear mixed model analysis. The outcome of the Bland Altman analysis shows the agreement of both algorithms with the force plate. However, with a linear mixed model analysis we were able to statistically test the effects of the independent variables, in this paper hoof, gait and number of analyzed trails. This test provides extra information about how the differences between the algorithms and the force plate arise. For this paper, we deemed this extra information important to able to give the reader an indication of how well the algorithms perform in different situations, i.e. different hooves and gaits. 

References:

1. Braganca FM, Bosch S, Voskamp JP, Marin-Perianu M, Van der Zwaag BJ, Vernooij JCM, et al. Validation of distal limb mounted inertial measurement unit sensors for stride detection in Warmblood horses at walk and trot. Equine Vet J. 2017;49(4):545-51.

2. Gunnarsson V, Tijssen M, Björnsdóttir S, Voskamp JP, Van Weeren PR, Back W, et al. Objective evaluation of stride parameters in the five-gaited Icelandic horse. 10th International Conference on Equine Exercise Physiology; 12 - 16 November 2018; Lorne, Australia2018. p. S52.

3. Serra Braganca FM, Tijssen M, Gunnarsson V, Björnsdóttir S, Persson-Sjodin E, Van Weeren PR, et al. The use of an artificial neural network to classify gait in Icelandic horses. 10th International Conference on Equine Exercise Physiology; 12 - 16 November 2018; Lorne, Australia2018. p. S41.

4. Starke SD, Clayton HM. A universal approach to determine footfall timings from kinematics of a single foot marker in hoofed animals. PeerJ. 2015;3:e783.

5. Tijssen M. Break-over detection using hoof-mounted inertial measurements units. Submitted to Journal of Experimental Biology. 2019b, companion paper.

6. Qualisys AB. Qualisys Track Manager - User Manual. Gothenborg, Sweden2011.

7. Hernlund E, Egenvall A, Roepstorff L. Kinematic characteristics of hoof landing in jumping horses at elite level. Equine Vet J Suppl. 2010(38):462-7.

---

## [Editor Report · Decision Letter 1]

4 May 2020

Automatic hoof-on and -off detection in horses using hoof-mounted inertial measurement unit sensors

PONE-D-19-27743R1

Dear Dr. Tijssen,

We are pleased to inform you that your manuscript has been judged scientifically suitable for publication and will be formally accepted for publication once it complies with all outstanding technical requirements.

With kind regards,

Chris Rogers

Academic Editor

PLOS ONE

Additional Editor Comments (optional):

Thank you for your thorough responses to the reviewers and edits to the manuscript. I am happy to say it may now proceed to publication.
---

## [Editor Report · Acceptance letter]

21 May 2020

PONE-D-19-27743R1 

Automatic hoof-on and -off detection in horses using hoof-mounted inertial measurement unit sensors

Dear Dr. Tijssen:

I am pleased to inform you that your manuscript has been deemed suitable for publication in PLOS ONE. Congratulations! Your manuscript is now with our production department. 

With kind regards,

on behalf of

Dr. Chris Rogers 

Academic Editor

PLOS ONE